# Nanobodies for the Early Detection of Ovarian Cancer

**DOI:** 10.3390/ijms232213687

**Published:** 2022-11-08

**Authors:** Lan-Huong Tran, Geert-Jan Graulus, Cécile Vincke, Natalia Smiejkowska, Anne Kindt, Nick Devoogdt, Serge Muyldermans, Peter Adriaensens, Wanda Guedens

**Affiliations:** 1Biomolecule Design Group, Institute for Materials Research (IMO-IMOMEC), Hasselt University, Agoralaan-Building D, BE-3590 Diepenbeek, Belgium; 2Laboratory of Cellular and Molecular Immunology, Vrije Universiteit Brussel, Pleinlaan 2, BE-1050 Brussels, Belgium; 3Laboratory of Medical Biochemistry, University of Antwerp, Prinsstraat 13, BE-2000 Antwerpen, Belgium; 4In Vivo Cellular and Molecular Imaging Laboratory (ICMI), Vrije Universiteit Brussel, Pleinlaan 2, BE-1050 Brussels, Belgium; 5Analytical and Circular Chemistry, Institute for Materials Research (IMO-IMOMEC), Hasselt University, Agoralaan-Building D, BE-3590 Diepenbeek, Belgium

**Keywords:** epithelial ovarian cancer, early-stage cancer detection, nanobodies, human epididymis protein 4, secretory leukocyte protease inhibitor, progranulin

## Abstract

Ovarian cancer ranks fifth in cancer-related deaths among women. Since ovarian cancer patients are often asymptomatic, most patients are diagnosed only at an advanced stage of disease. This results in a 5-year survival rate below 50%, which is in strong contrast to a survival rate as high as 94% if detected and treated at an early stage. Monitoring serum biomarkers offers new possibilities to diagnose ovarian cancer at an early stage. In this study, nanobodies targeting the ovarian cancer biomarkers human epididymis protein 4 (HE4), secretory leukocyte protease inhibitor (SLPI), and progranulin (PGRN) were evaluated regarding their expression levels in bacterial systems, epitope binning, and antigen-binding affinity by enzyme-linked immunosorbent assay and surface plasmon resonance. The selected nanobodies possess strong binding affinities for their cognate antigens (K_D_~0.1–10 nM) and therefore have a pronounced potential to detect ovarian cancer at an early stage. Moreover, it is of utmost importance that the limits of detection (LOD) for these biomarkers are in the pM range, implying high specificity and sensitivity, as demonstrated by values in human serum of 37 pM for HE4, 163 pM for SLPI, and 195 pM for PGRN. These nanobody candidates could thus pave the way towards multiplexed biosensors.

## 1. Introduction

Ovarian cancer (OC) is the fifth most deadly type of cancer in women [1,2,3,4,5,6]. The common symptoms of OC are indistinguishable and similar to other benign distress [4]. Most women are therefore diagnosed only at an advanced stage of the disease, resulting in a 5-year relative survival rate of around 47% [4,6,7,8,9,10,11]. Interestingly, epithelial ovarian cancer (EOC), which accounts for 90% of all OC cases, can be successfully treated, with a survival rate as high as 94%, if detected at an early stage. However, the early detection of EOC is difficult due to the lack of obvious symptoms or an effective screening test for dedicated biomarkers. Biomarkers are defined as cellular, biochemical, or molecular alterations that are measurable in biological media such as human tissues, cells, or fluids [12] and are being established as useful tools in diagnostics [13].

Several biomarkers, such as cancer antigen 125 (CA125) and human epididymis protein 4 (HE4) have been proposed for the diagnosis and therapy of ovarian carcinomas [4,7,9,14,15,16,17,18]. CA125 is a widely used cancer marker to monitor the response to treatment for a number of cancers, including EOC, and to detect the possible recurrence of cancer after treatment [3,19]. Conversely, HE4 has been uncovered more recently as a serum tumor marker for the early detection and monitoring of the recurrence or progression of the disease in patients with ovarian carcinomas [15,16,18]. HE4 belongs to the family of whey acidic four-disulfide core (WFDC) proteins (WAP). The combined monitoring of HE4, of which the normal value in healthy women ranges between 27.0 and 80.7 pM, and CA125 has been shown to be more sensitive in scoring OC than either marker separately [4]. Furthermore, secretory leukocyte protease inhibitor (SLPI) and progranulin (PGRN) are both overexpressed survival factors in EOC, of which the normal values in healthy woman range from 1923.1 to 3307.7 pM (25–43 ng/mL) and from 304.4 to 756.7 pM (27.4–67.2 ng/mL), respectively [6,20,21,22]. The occurrence of SLPI and PGRN proteins is correlated with cancer cell survival, proliferation, and invasion. As such, PGRN has been described as a prognostic biomarker for the advanced stages, while SLPI is considered an early detection marker of OC [6,10,14,17].

In order to detect these biomarkers for screening purposes, nanobodies (Nbs) offer several advantages over conventional antibodies, which are still widely used in diagnostic applications. Nbs, also referred to as VHHs, are single-domain antibodies derived from the variable domains of heavy-chain-only antibodies (Abs) circulating in the blood of camelids [23]. Because of their small size (12–15 kDa), superior biophysical stability, and antigen-binding properties, Nbs are considered as the next generation of antibodies, with great potential for medical applications, including therapy in humans [24,25,26]. Furthermore, Nbs are usually well expressed in bacteria, are stable monomeric fragments with good thermostability (up to 90 °C), and have high solubility, resistance to pH changes, and are encoded by a gene fragment of around 360–380 bp, with the latter being a property that allows them to be engineered easily [23,27,28,29,30]. In addition, Nbs offer great potential for a variety of applications such as biosensors as well as contrast probes for fluorescent or MRI imaging owing to their strong binding affinity and specificity towards their cognate antigens [31,32,33,34]. Nevertheless, the selectivity and sensitivity of the detection of OC biomarkers remains the limiting factor for disease detection at an early stage and for monitoring treatment efficiency and recurrence. In this paper, we present a selection of Nbs with high expression levels and superior selectivity and binding affinities for three different ovarian cancer biomarkers. Such nanobodies can pave the way towards more efficient biosensors for screening purposes and therapy follow-up, especially multiarray sensors that simultaneously monitor the CA125, HE4, SLPI, and PGRN biomarkers. 

## 2. Results and Discussion

### 2.1. Periplasmic Expression of the Nanobodies in E. coli WK6

The expression of the nanobodies (Nbs) was accomplished overnight at 28 °C in TB medium. The PelB leader sequence formed part of the expression vector and, when fused to the Nb, directed the Nb to the periplasm of the *E. coli WK6* cell, where PelB was cleaved off. The resulting nanobodies, with His_6_ tags at their C-termini, were subsequently purified from periplasmic extracts by IMAC followed by SEC. The Nbs were then evaluated for purity and molecular mass (15 kDa) by SDS-PAGE. A clear band was observed on the Coomassie-stained SDS-PAGE gels around 15 kDa, as shown in Appendix A. The expression level was defined as the amount (in mg) measured by OD at 280 nm of SEC-purified Nb per liter of culture (Table 1). The range of the expression levels of the various Nb clones could be quite large. The highest yields obtained for Nbs against HE4, SLPI, and PGRN were 9.1, 25.2, and 15.0 mg/L, respectively, while some other clones were produced at very low yields, well below 1 mg/L. 

### 2.2. Antigen-Binding Affinity of the Nanobodies

All purified Nbs were then evaluated for their antigen-binding capacity in an indirect ELISA where the antigen was coated on the plates. The ELISA results (Figure 1) show the binding profiles obtained by adding serially diluted purified Nbs to the wells. The EC50 values that correspond roughly to the equilibrium dissociation constants (K_D_) are given in Table 1. These estimated K_D_ values ranged from 4.6 to 746.8 nM, 2.0 to 91.5 nM, and 1.1 to 768.0 nM for the Nbs recognizing HE4, SLPI, and PGRN, respectively. Nbs with a k_off_ value that was too high for a proper curve fit are indicated with * in Table 1. Note that experiments with Nb clones with a K_D_ > 50 nM (as well as Nbs for which the K_D_ could not be determined from the experimental data) or an expression yield ≤ 0.25 mg/L were discontinued.

To confirm the ELISA results and to obtain the kinetic rate constants of the Nb-antigen binding (k_on_ and k_off_), an antigen-binding study was performed using SPR. The kinetic association rate (k_on_, expressed in M^−1^.s^−1^) and dissociation rate (k_off_, expressed in s^−1^) constants were determined, and the equilibrium dissociation constants (K_D_, expressed in nM) were calculated from the k_off_/k_on_ ratios. The SPR sensorgrams of some representative Nbs are shown in Figure 2, and the binding parameters can be found in Table 2. An overview of all SPR sensorgrams can be found in Appendix A, and the binding parameters are presented in Appendix A. The K_D_ values obtained by SPR and ELISA were in agreement. Promising Nb candidates for developing diagnostic tests to monitor the HE4, SLPI, and PGRN serum antigen concentrations were defined as those with a low K_D_ < 50 nM, a low k_off_ < 75 × 10^−3^ s^−1^, a high k_on_ of around 10^6^ M^−1^.s^−1^, and an expression yield > 0.25 mg/L. These Nbs have high antigen affinities as well as fast and strong antigen binding and are presented in Table 1.

To determine the limit of detection (LOD) for the antigens, two antibodies recognizing different independent epitopes (non-competitive) were required to perform a double-antibody sandwich ELISA test. Furthermore, Nb candidates for such Nb pairs should have low K_D_ values (SPR and ELISA) as well as high k_on_ and low k_off_ values (SPR), as seen for the Nbs listed in Table 1. It is further well-established that Nbs of the same family, which is the case for HE4-1 and HE4-6 (i.e., Nbs sharing a homologous CDR3 sequence of identical length), are by definition competitive, i.e., the recognize the same antigen epitope [35]. An overview of the Nb sequence families together with their antigen-binding properties and expression yields is shown in Appendix A. These Nbs were evaluated in epitope mapping experiments to determine the possible competition for the same antigen epitope. Remark that for Nbs of the same family, such as HE4-1 and HE4-6 of F3, similar antigen-binding properties can be assumed.

For the epitope mapping, the antigen was coated first on a CM5 chip through EDC/NHS chemistry. Then, the first Nb (Nb 1) was injected, resulting in a shift in the refractive index near the sensor surface (reported in response units (RU) along the y-axis) due to the binding of Nb 1 to the antigen. Once this reaction reached equilibrium, a mixture of the same Nb 1 and a second Nb (Nb 2) was injected. If the response increased further with the same amount as for Nb 2 alone, it indicated that Nb 2 showed no competition for binding to the same epitope, as it binds to another antigen epitope. If the increase in response was smaller or absent, it indicated (partial) competition between the two Nbs for the same antigen epitope. The experiment was subsequently repeated, but this time started with injecting Nb 2 alone, followed by the mixture of Nb 1 and Nb 2. Both curves should overlap to conclude that there was no competition for the same epitope. If this was not the case, it indicated that the binding of the first Nb affected the binding of the second Nb. The results are expressed relative to control experiments in which the same Nb was injected twice (same Nb for injection 1 and 2). To visually demonstrate the principle of epitope mapping, representative graphs are shown in Figure 3 for HE4 Nb pairs that are not competitive (HE4-1 vs. HE4-8), partially competitive (HE4-1 vs. HE4-4), and competitive (HE4-4 vs. HE4-11). HE4-1 is clearly not competitive with HE4-8 (Figure 3, top) because the response increased further with the same amount upon the addition of the mixture (purple arrow) as for HE4-1 (blue arrow) or HE4-8 (yellow arrow) alone. Conversely, when the increase in response was smaller, it meant that there was partial competition, as shown for HE4-1 with HE4-4 (Figure 3, middle). When there was no further increase in response upon adding the mixture, there was competition, as shown for HE4-4 with HE4-11 (Figure 3, bottom).

Epitope mapping experiments are shown in the supporting information (Appendix A). Taking into account that HE4-1 and HE4-6 have similar binding characteristics (same F3 family), non-competitive Nb pairs can be found such as HE4-1/HE4-8 (and therefore also HE4-6/HE4-8) and HE4-1/HE4-7 (see Table 2). In this study, the HE4-6/HE4-8 pair was used to determine the limit of detection of the HE4 antigen. Remark that HE4-8 has slightly better K_D_ and k_off_ values.

Regarding the SLPI antigen, Table 3 shows the non-competitive pairs. The SLPI-1/SLPI-7 pair was used to determine the limit of detection of the SLPI antigen. Remark that SLPI-3 has a high K_D_ value and SLPI-9 has a very low expression yield.

Regarding the PGRN antigen, Table 4 shows the non-competitive pairs. The PGRN-5/PGRN-7 pair was used to determine the limit of detection of the PGRN antigen. Remark that PGRN-7 has slightly better K_D_ and k_off_ values compared to PGRN-8, PGRN-9, and PGRN-11, but it is clear that other non-competitive pairs would have been possible. 

### 2.3. Limits of Detection of Antigens

The detection limits of the selected non-competitive Nb pairs for their respective antigens were derived from ELISA, as shown in Figure 4 and Table 5 (the 450 nm absorption data and calculations can be found in Appendix A). The data were analyzed, and the LOD values were calculated as the sum of the mean of the blank measures and three times the standard deviation of the blank measures. The LOD values for SLPI (163 pM in human serum) and PGRN (195 pM in human serum) were much lower than the concentrations of these antigens in the serum of healthy women (i.e., 1923.1–3307.7 and 304.4–746.7 pM for SLPI and PGRN, respectively). Meeter et al. previously determined PGRN levels as low as 5.2 ng/mL (pM) using a commercially available ELISA kit [36]. The LOD value of 37 pM for HE4 in human serum lies in the same order of magnitude as the concentration levels in healthy women (27.0 to 80.7 pM). A previous study by Huhtinen et al. reported significantly elevated HE4 levels (mean value of 1125.4 pM) in patients with ovarian malignancies [37]. Although early-stage values will be lower, the determined serum LOD of 37 pM renders the selected nanobody suitable for detecting significant increases in the HE4 antigen above the upper limit of 80.7 pM that was reported for healthy women [37]. Moreover, the LOD values obtained in this study were significantly lower than the observed mean values for HE4 (four times lower). The LOD values were 15 times lower for SLPI and approximately 3.5 times lower for PGRN [1,3,22,37]. These results indicate a high sensitivity of the described nanobodies for these antigens, which could pave the way towards the diagnosis of OC at an early stage.

## 3. Materials and Methods

### 3.1. Materials

pGEX-HE4 was a gift from prof. dr. Ronny Drapkin (https://www.addgene.org/18100/) (accessed on 4 October 2022). HisPur Ni-NTA Resin and Clear polystyrene Nunc F Maxisorp 96-well plates were purchased from Thermo Fisher Scientific (Merelbeke, Belgium). Anti-Myc and anti-HA antibodies were purchased from Biolegend (San Diego, CA, USA); Anti-His antibody was purchased from AbD Serotec (Kidlington, UK); Anti-mouse-IgG alkaline phosphate (AP) was purchased from Bioké (Leiden, The Netherlands); and an AP substrate was purchased from Sigma-Aldrich (Overijse, Belgium). Human WAP5/WDFC2/HE4 protein (His tag) antigen was purchased from Bioconnect (Huissen, The Netherlands), and recombinant human SLPI and PGRN antigens were purchased from R&D systems (Abington, UK). Buffers and growth media were prepared using chemicals from Merck: LB medium (10 g/L Bacto-tryptone, 5 g/L Bacto-yeast extract, and 10 g/L NaCl); LB agar (LB medium with 15 g/L Agar); and TB medium (1 L: 24 g of yeast extract, 12 g of tryptone, 4 mL of glycerol, and 100 mL of 0.17 M KH_2_PO_4_ buffer.

### 3.2. Methods

#### 3.2.1. Generation of Nanobodies

Nanobodies were generated using published protocols [29] by immunizing dromedaries with recombinant proteins, extracting RNA from peripheral blood lymphocytes, amplifying the nanobody-coding sequences by two-step RT-PCR, and cloning the pHEN4 phagemid vector. Nanobodies were phage-displayed, and several rounds of biopanning were performed on recombinant proteins that were immobilized on immunosorbent plastic. Bacterial periplasmic extracts of individual nanobody clones were tested in ELISA for specific binding to the recombinant proteins, and positive hits were sequenced. Nanobodies were then recloned in pHEN6c, pMECS, or pHEN18 bacterial expression plasmids, coding for nanobodies with, respectively, carboxyterminal hexahistidine, HA- and hexahistidine, or c-Myc and hexahistidine tags.

The recombinant proteins for immunization, panning, and ELISA screenings were as follows: SLPI (RnD Systems catalog number 1274-PI), HE4-GST fusion protein (produced and purified from Sf9 insect cells; kindly provided by Dr. Ronny Drapkin, Dana Farber Institute, Boston, MA, USA), and progranulin (produced and purified from mammalian cells; kindly provided by Dr. Renato Iozzo, Thomas Jefferson University, Philadelphia, PA, USA).

#### 3.2.2. Expression and Periplasmic Extraction of Nbs in *E. coli WK6*

An expression vector (pHEN18 for HE4, pHEN6c for SLPI, and pMECS for PGRN Nbs), including the nanobody (Nb) gene with a *pelB* leader sequence and hexa-histidine tag (His_6_ tag) at the N- and C-terminals of the Nb, respectively, was transformed in *E. coli WK6* and plated on LB agar dishes complemented with 100 µg/mL ampicillin. The Nb was obtained by starting from a single colony and culturing at 37 °C in baffled shake flasks containing TB medium with ampicillin at 100 µg/mL. The expression of the Nb was induced by adding isopropyl β-D-1-thiogalactopyranoside (IPTG) to a final concentration of 1 mM, and the culture was shaken overnight at 28 °C. The Nb was directed to the periplasmic space of *E. coli* via the PelB sequence, which was removed by a signal peptidase. Afterwards, the cell culture was harvested, and the cell pellet was obtained by centrifuging at 7000× *g* for 8 min at 14 °C. The Nbs were extracted from the periplasm by an osmotic shock in TES buffer (0.2 M Tris-Cl pH 8.0; 0.5 mM EDTA disodium dihydrate; 0.5 M Sucrose). The cellular pellets obtained from 330 mL of cell culture were resuspended in 4 mL of TES buffer and pipetted up and down rigorously until the suspensions were free of clumps. The suspensions were shaken gently for 6 h on ice in a cold room. The *E. coli* cultures were given an osmotic shock by adding 8 mL of 1:4 diluted TES buffer (TES/4) per pellet. The cell suspensions were further incubated overnight under the same conditions. The periplasmic protein extracts were harvested by centrifuging the cell suspension at 7000× *g* for 30 min at 14 °C.

#### 3.2.3. Nb Protein Purification

The His_6_-tagged Nb was purified from the periplasmic extract by immobilized metal affinity chromatography (IMAC) followed by size-exclusion chromatography (SEC) and analyzed for purity and molecular weight by SDS-PAGE. For the IMAC purification, the HisPur Ni-NTA resin was washed with PBS, followed by centrifuging at 1400 rpm for 7 min and the decantation of the supernatant. This was repeated twice, after which the slurry was swirled rigorously, to resuspend the beads, and added to the periplasmic extract. The mixture was then incubated for 1 h on a shaking platform at 100 rpm, followed by centrifuging at 1400× *g* rpm for 7 min. Part of the supernatant was decanted until the volume was reduced by 50%. The remaining part was swirled and loaded on a PD-10 column (Cytiva, Hoegaarden, Belgium), followed by washing with 20 mL of PBS per mL of HIS-Select suspension. The captured His_6_-tagged Nbs were subsequently eluted with 5 × 1 mL of PBS containing 0.5 M imidazole and were collected in fractions for concentration measurements at 280 nm using a Nanodrop 1000 spectrophotometer (Thermo Fisher Scientific (Merelbeke, Belgium). 

Further purification of the Nbs was then performed by SEC on an ÄKTAxpress using a Hiload Superdex 75 HR 16/60 column (Thermo Fisher Scientific (Merelbeke, Belgium) equilibrated with PBS. PBS (pH 7.4) was used as a running buffer at a flow rate of 0.5 mL/min. Fractions showing significant UV absorption at 280 nM were pooled, and the protein concentration was measured using a Nanodrop 1000 spectrophotometer. The purities and molecular weights of the Nbs were checked by SDS-PAGE and Coomassie staining.

#### 3.2.4. Enzyme-Linked Immunosorbent Assay

The antigen-binding capacities of the purified Nbs were tested via indirect ELISA. An antigen (at a concentration of 1 µg/mL) was first coated overnight at 4 °C in wells of 96-well ELISA plates, and the remaining binding sites were blocked for 2 h at room temperature with 2% skim milk, followed by the addition of ¼ serial dilutions of purified Nb (dilutions ranging from 2 µM to 7.5 pM). PBS was used as a control. Antigen-bound Nbs were detected using sequential additions of mouse anti-Myc (for HE4), mouse anti-His (for SLPI), and mouse anti-HA (for PGRN) as the primary antibodies (final concentration of 1 µg/mL) and goat anti-mouse IgG-conjugated alkaline phosphatase as a secondary antibody (final concentration at 1 µg/mL). Washing with PBS containing 0.05% Tween 20 was performed between each step. The absorbance was measured at 405 nm after the addition of the AP substrate at a concentration of 2 mg/mL in AP buffer every 5 min while the plates were kept at room temperature.

#### 3.2.5. Surface Plasmon Resonance and Epitope Binning

The interaction kinetics and affinities of the Nbs for their antigens were measured by surface plasmon resonance (SPR) on a Biacore T200 (Cytiva, Hoegaarden, Belgium) instrument.

For SPR, the antigen was coated on a CM5 chip by EDC/NHS (1-ethyl-3-(3-dimethylaminopropyl)carbodiimide/N-hydroxysulfosuccinimide) chemistry. A kinetic study was performed, as described by the manufacturer’s protocol, using a two-fold serial dilution of Nbs. Depending on the K_D_ value obtained in the ELISA experiments, the Nb concentration ranges used were 500–1.95 nM (for high K_D_ values, >10 nM) or 125–0.49 (for low K_D_ values, <10 nM). A flow rate of 30 µL/min in HBS buffer was used, combined with an association phase of 120 s and a dissociation phase of 600 s. After running, 100 mM glycine-HCl (pH 2.5) was used for regeneration. The results were then fitted using a 1:1 Langmuir binding with a drift component in an RI2 kinetic model (BIA evaluation software of Biacore T200).

The epitope binning was also performed on the Biacore T200 using the same antigen-coated CM5 chip. The antigen was saturated by flowing the first Nb (at a concentration of 100× its K_D_ value) for 300 s at 10 µL/min. This was followed by a mixture of the same concentration of the first Nb and a second Nb for another 300 s (also at a concentration of 100× its K_D_ value) and a dissociation phase of 600 s. The chip was subsequently regenerated with 100 mM glycine-HCl (pH 2.5) for 30 s at a flow rate of 30 µL/min.

#### 3.2.6. Determination of the Limit of Detection

The limit of detection (LOD) is defined as the lowest concentration of an analyte in a sample that can be detected. The LOD is given as the sum of the mean of the blank measures and three times the standard deviation of the blank measures. To determine the lowest concentration of antigen that can be detected, a sandwich ELISA was performed.

A pair of non-competitive Nbs (binding to different antigen epitopes) was selected for the HE4, SLPI, and PRGN antigens, which had low K_D_ values and low k_off_ values, i.e., HE4-6 and HE4-8, SLPI-1 and SLPI-7, and PGRN-5 and PGRN-7. HE4-8, SLPI-7, and PGRN-7 were used as the antigen-capturing Nbs, and HE4-6, SLPI-1, and PGRN-5 were used as the detecting Nbs in the sandwich ELISA. These Nbs were first recloned into a pHEN6c plasmid with an LPETGG tag at the C-terminus for the purpose of further biotinylation via the Sortase A protein ligation (SPL) technique [38]. The Nbs were then biotinylated via the SPL technique. The details of recloning and biotinylation via SPL can be found in Appendix A. The sandwich ELISA was performed as follows:

The first Nb at 0.1 μM was first coated overnight on 96-well Nunc F Maxisorp ELISA plates at 4 °C, followed by the addition of 200 µL of protein-free blocking buffer (PBS) and incubation at 37 °C for 1 h. Antigens from ½ serial dilutions in PBS or human serum were added (Table 6), and incubation took place for 1 h at room temperature. The second, biotinylated Nb at 0.5 μM was then added. Wells without antigen were used as controls. Bound biotinylated Nbs were detected using a sequential addition of Streptavidin-HRP (diluted 20,000 times to a final concentration of 55 ng/mL). Between the consecutive steps, the wells were washed with PBS containing 0.05% Tween 20. The color development was started by the addition of 100 µL of HRP substrate (1-Step Ultra TMB-ELISA), and the reaction was stopped by adding 100 µL of a 2 M H_2_SO_4_ solution before the measurement of the absorbance at 450 nm. All measurements were performed in quadruplicate, and the data were analyzed with excel and Graphpad Prism 8 software for statistical analysis to derive the LOD values.

## 4. Conclusions

In this study, nanobodies targeting human HE4, SLPI, and PGRN ovarian cancer antigens were explored with respect to their bacterial expression yield, antigen-binding affinity, and antigen competition. For each of these antigens, the best-performing nanobodies were selected, and non-competitive pairs were used in a sandwich ELISA to determine the LOD. The excellent LOD values obtained for all three antigens in human serum indicated the very specific and sensitive binding of the antigens to the selected Nbs. The LOD values in human serum obtained for HE4, SLPI, and PGRN were 37, 163, and 195 pM, respectively. These LOD values were significantly lower than the concentrations of these antigens in the serum of healthy women, i.e., 150, 2462, and 656 pM for HE4, SLPI, and PGRN, respectively.

The presented results also indicate that the selected nanobodies might be excellent candidates for the early diagnosis of EOC as well as for the development of multiarray biosensors in which multiple relevant biomarker antigens are detected simultaneously in a highly sensitive and selective manner. Such applications would indeed benefit from the nanobodies’ increased stability in terms of pH and temperature, small size, and ease of modification compared to conventional antibodies. The latter properties are valuable when considering multiplexed sensors that contain multiple immobilized sensing elements on a single transducing element. Future work will need to validate if sensing formats that rely on the proposed nanobodies are indeed capable of discriminating between EOC patients and healthy women.

As demonstrated in the recent literature, similar attempts describe the great potential of Nbs for diagnosing several other human cancers, as shown in the following overview (Table 7).

## Figures and Tables

**Figure 1 ijms-23-13687-f001:**
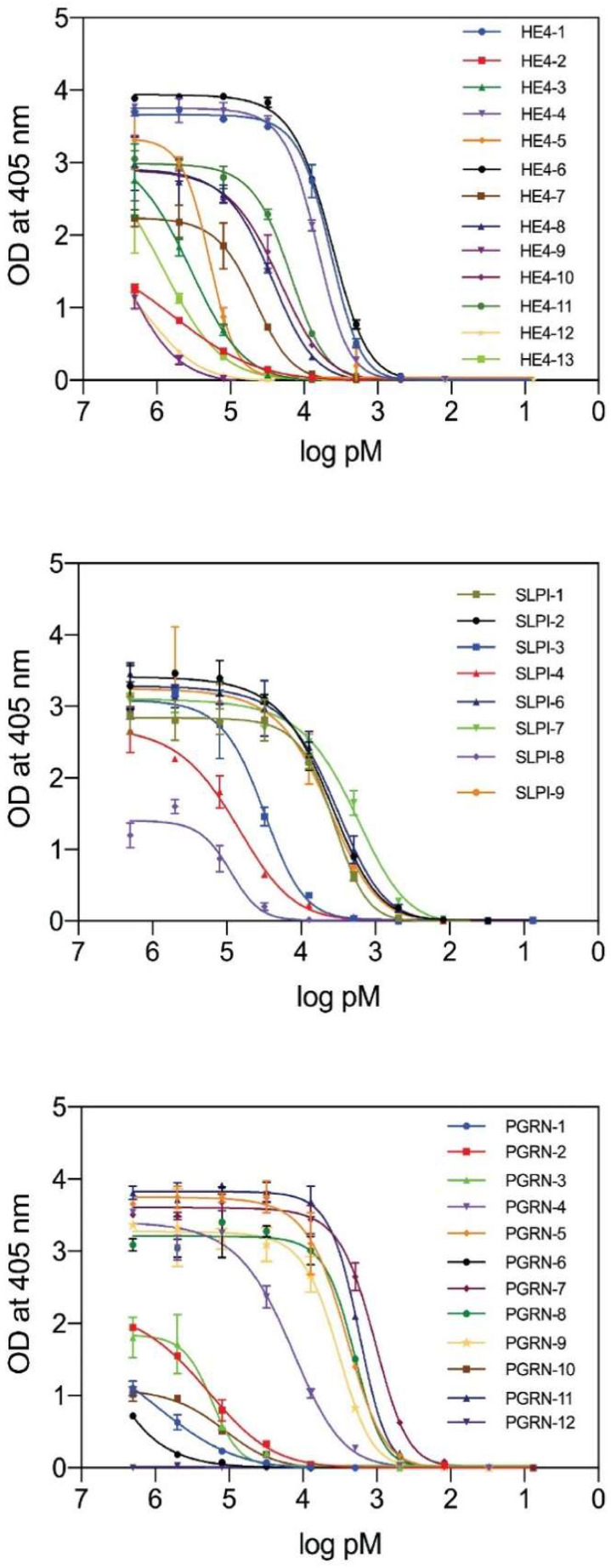
Indirect ELISA results showing the antigen-binding capacities of the nanobody variants targeting HE4 (**top**), SLPI (**middle**), and PGRN (**bottom**).

**Figure 2 ijms-23-13687-f002:**
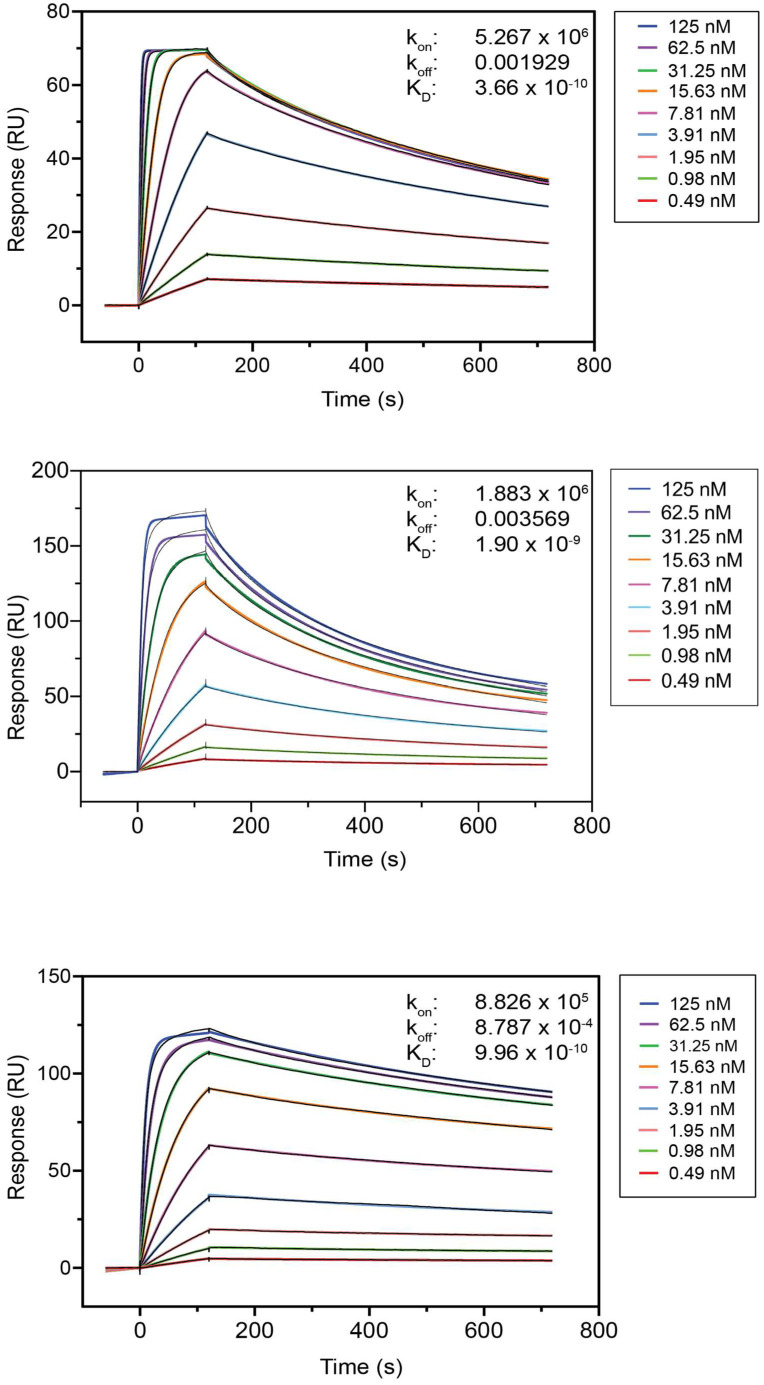
Antigen-binding affinity of Nbs HE4-1 (**top**), SLPI-7 (**middle**), and PGRN-7 (**bottom**), as determined by SPR (k_on_ in M^−1^.s^−1^ and k_off_ in s^−1^).

**Figure 3 ijms-23-13687-f003:**
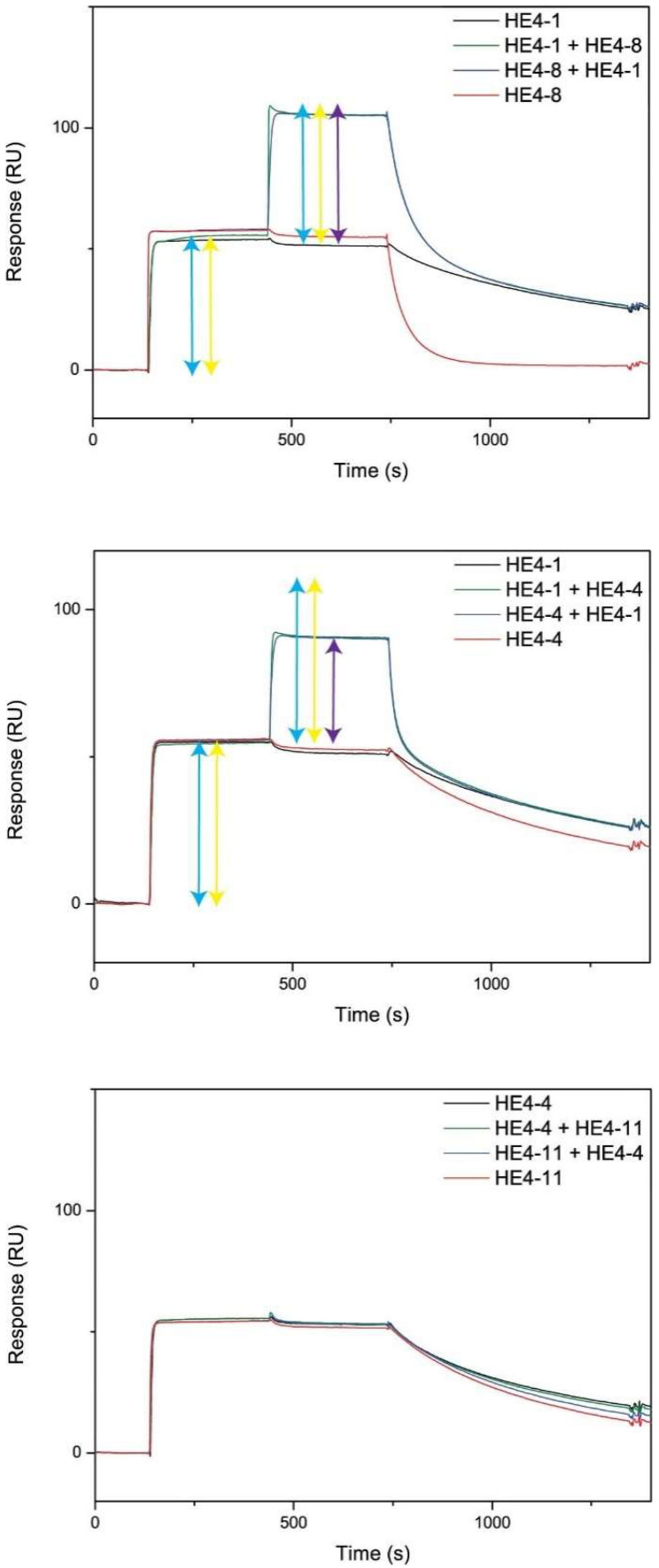
Competition in the antigen binding of HE4 Nbs as determined by epitope mapping. HE4-1 with HE4-8 (not competitive, **top**), HE4-1 with HE4-4 (partially competitive, **middle**), and HE4-4 with HE4-11 (competitive, **bottom**).

**Figure 4 ijms-23-13687-f004:**
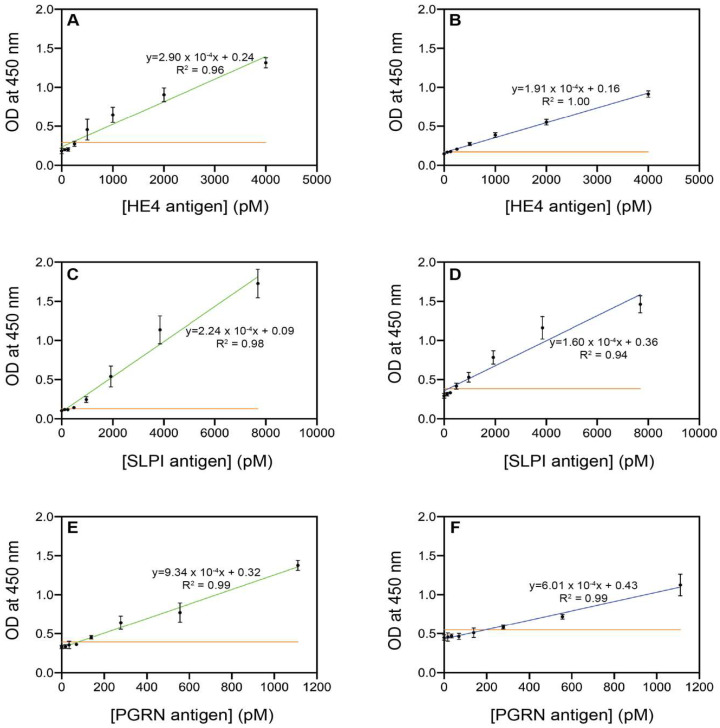
Sandwich ELISA plots showing the detection results of the recombinant human antigens HE4, SLPI, and PGRN by their respective best-performing nanobodies: HE4-6 in PBS (**A**) and serum (**B**); SLPI-1 in PBS (**C**) and serum (**D**); and PGRN-5 in PBS (**E**) and serum (**F**). The yellow line indicates the LOD.

**Table 1 ijms-23-13687-t001:** Expression levels of the nanobodies targeting HE4, SLPI, and PGRN and equilibrium dissociation constants (K_D_), as measured by indirect ELISA and expressed by the EC50 value. Clones expressed at levels >25 mg/L and showing a K_D_ < 50 nM were further studied using SPR to assess their binding affinity.

Targeted Antigen	Nb Clone	Expression Level (mg/L)	EC50 ^a^ (nM)	k_on_ ^b^ (M^−1^.s^−1^)	k_off_ ^b^ (s^−1^)	K_D_ ^b^ (nM)
HE4	HE4-1	4.4	4.6	5.3 × 10^6^	1.9 × 10^−3^	0.4
	HE4-2	0.6	509.7	-	-	-
	HE4-3	1.7	353.9	-	-	-
	HE4-4	0.2	6.8	-	-	-
	HE4-5	2.2	198.1	-	-	-
	HE4-6	7.7	4.6	3.2 × 10^6^	9.0 × 10^−3^	2.8
	HE4-7	5.9	48.2	1.1 × 10^6^	36.3 × 10^−3^	32.0
	HE4-8	4.8	28.7	1.8 × 10^6^	28.9 × 10^−3^	16.0
	HE4-9	1.9	*	-	-	-
	HE4-10	2.1	23.3	2.0 × 10^6^	34.2 × 10^−3^	18.0
	HE4-11	7.4	16.0	0.4 × 10^6^	2.6 × 10^−3^	6.1
	HE4-12	3.3	*	-	-	-
	HE4-13	9.1	746.8	-	-	-
SLPI	SLPI-1	20.9	4.1	1.3 × 10^6^	56.4 × 10^−3^	43.0
	SLPI-2	0.2	4.3	-	-	-
	SLPI-3	2.5	32.4	-	-	-
	SLPI-4	0.8	78.9	-	-	-
	SLPI-6	25.2	3.4	-	-	-
	SLPI-7	0.3	2.0	1.9 × 10^6^	3.6 × 10^−3^	1.9
	SLPI-8	1.3	91.5	-	-	-
	SLPI-9	0.1	4.0	-	-	-
PGRN	PGRN-1	10.6	768.0	-	-	-
	PGRN-2	10.0	198.0	-	-	-
	PGRN-3	1.3	175.0	-	-	-
	PGRN-4	15.0	15.0	-	-	-
	PGRN-5	10.5	2.7	0.1 × 10^6^	3.4 × 10^−3^	33.0
	PGRN-6	9.6	*	-	-	-
	PGRN-7	0.5	1.1	0.9 × 10^6^	0.9 × 10^−3^	1.0
	PGRN-8	3.3	2.1	0.5 × 10^6^	9.1 × 10^−3^	17.0
	PGRN-9	1.5	3.5	1.9 × 10^6^	57.2 × 10^−3^	30.0
	PGRN-10	14.0	122.0	-	-	-
	PGRN-11	11.3	1.8	0.3 × 10^6^	2.0 × 10^−3^	7.5
	PGRN-12	0.9	*	-	-	-

^a^ Obtained via ELISA; ^b^ obtained via SPR for selected Nb clones; * = could not be determined from the experimental data.

**Table 2 ijms-23-13687-t002:** Non-competitive Nb pair possibilities for HE4 (C—competitive; P.C—partially competitive; N.C—non-competitive).

	HE4-1/HE4-6	HE4-4	HE4-7	HE4-8	HE4-11
HE4-1/HE4-6	C	P.C	N.C	N.C	P.C
HE4-4		C	C	C	C
HE4-7			C	C	C
HE4-8				C	C
HE4-11					C

**Table 3 ijms-23-13687-t003:** Non-competitive Nb pair possibilities for SLPI (C—competitive; P.C—partially competitive; N.C—non-competitive).

	SLPI-1	SLPI-2	SLPI-3	SLPI-7	SLPI-9
SLPI-1	C	P.C	P.C	N.C	P.C
SLPI-2		C	N.C	P.C	C
SLPI-3			C	N.C	N.C
SLPI-7				C	P.C
SLPI-9					C

**Table 4 ijms-23-13687-t004:** Non-competitive Nb pair possibilities for PGRN (C—competitive; P.C—partially competitive; N.C—non-competitive).

	PGRN-5	PGRN-7	PGRN-8	PGRN-9	PGRN-11
PGRN-5	C	N.C	N.C	N.C	C
PGRN-7		C	N.C	C	N.C
PGRN-8			C	N.C	N.C
PGRN-9				C	N.C
PGRN-11					C

**Table 5 ijms-23-13687-t005:** Comparison between the serum levels in healthy women and the obtained limits of detection (LOD) for the recombinant human HE4, SLPI, and PGRN antigens by the selected nanobodies HE4-6, SLPI-1, and PGRN-5, respectively.

	HE4	SLPI ^a^	PGRN
Serum level in healthy women (pM)	27.0–80.7	1923.1–3307.7	304.4–746.7
Reference	[38]	[22]	[6]
LOD in PBS (pM), this study	187	174	76
LOD in human serum (pM), this study	37	163	195

^a^ For SLPI, the cited reference reports the interquartile interval for the antigen in healthy women.

**Table 6 ijms-23-13687-t006:** Antigen dilution series for each Nb (dilutions in PBS and human serum).

HE4 Concentration	SLPI Concentration	PGRN Concentration
100 ng/mL (4000 pM)	100 ng/mL (7692.3 pM)	100 ng/mL (1111.1 pM)
50 ng/mL (2000 pM)	50 ng/mL (3846.1 pM)	50 ng/mL (555.5 pM)
25 ng/mL (1000 pM)	25 ng/mL (1923.1 pM)	25 ng/mL (277.8 pM)
12.5 ng/mL (500 pM)	12.5 ng/mL (961.5 pM)	12.5 ng/mL (138.9 pM)
6.25 ng/mL (250 pM)	6.25 ng/mL (480.8 pM)	6.25 ng/mL (69.4 pM)
3.12 ng/mL (125 pM)	3.12 ng/mL (240.4 pM)	3.12 ng/mL (34.7 pM)
1.56 ng/mL (62.5 pM)	1.56 ng/mL (120.2 pM)	1.56 ng/mL (17.4 pM)
0 ng/mL (0 pM)	0 ng/mL (0 pM)	0 ng/mL (0 pM)

**Table 7 ijms-23-13687-t007:** Nanobodies used for cancer diagnostics.

Nanobody	Target	Application	Reference
VHH; ^99m^Tc-PSMA6; ^99m^Tc-PSM30; JVZ-007	PSA, PSMA	Prostate cancer	[39,40,41]
cABPSA-N7; cAbPSA-C23	PSA	Prostate cancer	[40]
Radiolabeled Nb	HER2	Breast cancer	[42]
^99m^Tc-labeled Nb	EGFR	Epithelial cell tumors (skin, lung, head, and neck tumors)	[43,44]
NIR dye-coupled Nb	HER-2	Cancer imaging	[43]
NJB2	ECM	Primary tumor detection	[45]
sdAb-HER2-QD; sdAB-CEA-QD	HER2, CEA	Detection and imaging of human micrometastases	[46]
VHH	Alpha-fetoprotein	Cancer biomarker for liver cancer	[39]
Biotin-NB29	FGL1	Cancer immunotherapy	[47]
^99m^Tc-NM-02	HER2	Breast cancer	[48]
^68^GaNOTA-Anti-HER2	HER-2	Breast cancer	[42,48,49]
MSB0010853	HER3	Non-small-cell lung cancer and head and neck cancer	[41]
^99m^Tc-labeled Nb	EGFR	Detection of tumor cells and lung and head cancers	[50]
D10	EGFR	Human epidermoid carcinoma	[51]
2Rs15d	HER-2	Breast cancer	[44,51]
^131^I-SGMIB-Anti-HER2	HER-2	Breast cancer	[49,52,53]
Quantum dot Nb	EGF	Detection of cancer	[54]
^99m^Tc-Anti-PD-L1	PD-L1	Non-small-cell lung cancer	[49]
TAS266	DR5	Advanced solid tumors	[49]

## Data Availability

The data presented in this study are available on request from the corresponding author. The data are not publicly available due to intellectual property agreements between the collaborating research groups.

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
