# Peer review of "Nanobodies for the Early Detection of Ovarian Cancer"

_ijms, 2022, doi:10.3390/ijms232213687_

Round 1

Reviewer 1 Report

This is a relatively complex paper describing the laboratory development of nanoantibodies aiming at the early detection of ovarian cancer. Some parts of the materials and methods sections (eg Materials 2.1) could be written in a more simplified manner. Also, in the conclusion section it would be of interest to the reader to report any similar attempts undergoing in the early detection of other human cancers.  

Author Response

Thank you very much for your constructive remarks. A comparison overview is implemented in the conclusion listing similar nanobodies for the diagnosis of other types of human cancers as proposed by reviewer 1. Two lines are added on top of the overview to situate. The Materials and Methods section has been rewritten in a simplified manner, especially the Materials (2.1) and Methods section 2.2.3 “Nb protein purification”.

Reviewer 2 Report

The article submitted by Lan-Huong Tran et al. reported Nanobodies for the early detection of ovarian cancer. They showcase three different ovarian cancer biomarkers for which a particular set of Nbs has been found to have very high expression levels, binding selectivity, and binding affinity. After immunizing dromedaries with recombinant proteins, RNA was extracted from peripheral blood cells. The coding sequences for the nanobodies were amplified using two-step RT-PCR and cloned into the pHEN4 phagemid vector, all according to established techniques. In a single experiment, these nanobodies simultaneously screened four cancer biomarkers: CA125, HE4, SLPI, and PGRN. The article is interesting, and the described work has novelty. It can be published after minor revisions. The authors should try to give a comparison table to summarize similar nanobodies in the literature.

Author Response

Thank you very much for your constructive comments. A comparison overview is implemented in the conclusion listing similar nanobodies for the diagnosis of other types of human cancers as proposed by reviewer 2. Two lines are added on top of the overview to situate.